# A Rare Single Case of COVID-19-Induced Acute Myocarditis and Encephalopathy Presenting Simultaneously

**DOI:** 10.3390/vaccines11030541

**Published:** 2023-02-24

**Authors:** Usman Saeedullah, Anas M. Abbas, Caitlin Ward, Maha Bayya, Jenish Bhandari, Araf M. Abbas, Joshua DeLeon, Allison B. Reiss

**Affiliations:** 1Department of Medicine and Biomedical Research Institute, NYU Long Island School of Medicine, 101 Mineola Boulevard, Suite 4-004, Mineola, NY 11501, USA; 2SUNY Upstate Medical University, 750 E Adams St, Syracuse, NY 13210, USA; 3College of Health Professions & Human Services, Department of Population Health, Hofstra University 1000 Hempstead Turnpike, Hempstead, NY 11549, USA

**Keywords:** myocarditis, encephalopathy, COVID-19, ACE-2, dilated cardiomyopathy, Takutsobo

## Abstract

The ongoing coronavirus disease 2019 (COVID-19) pandemic may result in cardiovascular complications such as myocarditis, while encephalitis is a potentially life-threatening COVID-19-associated central nervous system complication. This case illustrates the possibility of developing severe multisystem symptoms from a COVID-19 infection, despite having received the COVID-19 vaccine within the year. Delay in treatment for myocarditis and encephalopathy can lead to permanent and possibly fatal damage. Our patient, a middle-aged female with a complicated medical history, initially came in without characteristic manifestations of myocarditis such as shortness of breath, chest pain, or arrhythmia, but with an altered mental status. Through further laboratory tests, the patient was diagnosed with myocarditis and encephalopathy, which were resolved within weeks through medical management and physical/occupational therapy. This case presentation describes the first reported case of concomitant COVID-19 myocarditis and encephalitis after receiving a booster dose within the year.

## 1. Introduction

The severe acute respiratory syndrome coronavirus 2 (SARS-CoV-2), responsible for coronavirus disease 2019 (COVID-19), caused an outbreak that has spread across the globe, leading to massive infection, morbidity and mortality [1]. Overall, 80.8% of the US population has received at least one dose of an approved vaccine, with 69% completing the primary series [1]. COVID-19 vaccines have proven clinical efficacy, inducing an antibody response, lowering the risk of SARS-CoV-2 infection and reducing mortality if infection occurs [2]. However, vaccine effectiveness against emergency department/urgent care visits dropped from 87% 2 months after a booster to 66% 4–5 months later. A similar drop in vaccine effectiveness was found in the ability to protect against hospitalizations, with a decrease from 91% up to 2 months after a booster to 78% after 4 months [3]. Waning levels of antibodies pave the way for reinfection and susceptibility to different strains of COVID-19. Variants of SARS-CoV-2 have been circulating since the beginning of the pandemic, leading to constant outbreaks and questions about vaccine efficacy [4].

A recent meta-analysis estimated the global prevalence of post-COVID-19 at 0.43 and 0.31 for North America [5]. Prevalence of hospitalization was calculated to be 0.54 [5]. A cohort study found that dyspnea was the most prevalent complaint during hospitalization [6]. Over the course of the pandemic, reports of multi-organ system involvement during infection have increased, including the cardiovascular and central nervous system (CNS) [7,8,9,10,11,12,13,14,15,16].

At first recognized as a respiratory disease, COVID-19 has been associated with cardiac dysfunction even in the absence of respiratory manifestations [7]. Chronic comorbidities such as hypertension have been associated with the severity of cardiac illness [8]. The mechanisms of cardiac involvement during a COVID-19 infection are not well understood, with studies attributing cardiac injury to the direct SARS-CoV-2 infection of cardiac cells, or myocardial damage developing from an increased metabolic demand due to a systemic infection and ongoing hypoxia [9,10,11]. Several case series linked elevated cardiac biomarkers to myocardial injury, leading to a poorer prognosis and increased mortality rate [10]. Myocarditis is a common cardiac presentation associated with COVID-19 infection. Usually diagnosed through electrocardiogram (ECG) and echocardiogram, myocarditis can be suspected clinically but absent in ECG and echocardiogram [12]. Differential diagnosis and treatment should consider cardiac injury along with identifying and treating underlying comorbidities.

On top of possible respiratory and cardiac involvement, neurologic symptoms have been reported in up to 36% of patients hospitalized with COVID-19 [13]. Commonly reported CNS symptoms of COVID-19 may include headache, dizziness and memory deficits [14]. A systematic review found up to 23% of patients who were diagnosed with encephalopathy, which is associated with more severe COVID-19 [14,15]. Common symptoms of patients with COVID-19-induced encephalopathy were disorientation or confusion, decreased consciousness, seizures and headaches [16]. Patients with COVID-19-induced encephalopathy presented with upper respiratory tract infection, altered mental status and rapid respiratory failure [16]. Research found that encephalopathy from COVID-19 had a greater effect on the cortex than the meninges as opposed to herpes simplex virus (HSV) encephalitis [16]. Diffusion to the CNS currently remains unclear, but there are proposed mechanisms of neurotropism [17]. There are various methods used in diagnosing viral encephalopathy, including magnetic resonance imaging (MRI) and electroencephalography (EEG) [18]. These testing methods can detect the differences between acute, focal and generalized encephalopathy, as well as the type of the condition [18]. Treatment for COVID-19-induced encephalopathy has not been well studied, but the accepted treatment for viral encephalopathy has been acyclovir secondary to the common HSV encephalitis [16,18]. Corticosteroids, immunomodulators or immunoglobulins may also be employed [19,20]. Vaccination is recommended in preventing viral encephalopathy [18].

After an extensive literature review, we believe that this case presentation describes the first reported case of concomitant COVID-19 myocarditis and encephalitis after receiving a booster dose within the year.

## 2. Case Presentation

The patient is a 54-year-old female with a past medical history of lumbar radiculopathy, gastro-esophageal reflux disease, post-concussion syndrome and intractable nausea and vomiting who initially presented to Upstate University Hospital (UUH) because of failure to thrive and recurrent episodes of syncope. Upon presentation to the emergency department at UUH, the patient underwent a CT without contrast of the abdomen and pelvis, which showed multiple small blastic lesions throughout the bones, and lytic lesions in several lumbar vertebrae. A nodular lesion in the left breast was also found, which raised concern for metastatic breast cancer. The oncology team was consulted and recommended obtaining values for CA 15-3, CA 19-9, CA 27.29 and CEA tumor markers, all of which were within the normal range (Table 1). The breast surgery team was consulted and advised that the patient should follow up on the breast mass on an outpatient basis. The patient was not started on chemotherapy or immunosuppressants during this hospitalization.

The patient originally tested positive for COVID-19 on 10 October 2022, when she presented to an outside hospital for syncopal episodes. The patient had received three total Pfizer COVID-19 vaccinations, with her most recent booster vaccination in December 2021. Upon arrival at UUH, the patient had no symptoms of COVID-19, and therefore received no treatment for COVID-19 while in the hospital. An antibody titer for COVID-19 was run on the patient to see if lasting immunity still existed from the previous booster shot received by the patient a year ago. ELISA lab titration performed at UUH for COVID-19 antibodies showed waning immunity in the patient from prior combinations of infection and vaccination. Titer values received were at 1:100, which falls in the low range of antibody detection.

While at the outside hospital on 10 October 2022, she received an echocardiogram as part of her syncope workup, which showed a normal left ventricle with an ejection fraction of 55–60% and normal systolic and diastolic function. A repeat echocardiogram was ordered on 26 October 2022 while the patient was at UUH, which showed an ejection fraction of 31% with wall motion abnormalities (Figure 1, Table 2). ProBNP levels were elevated at 7616 pg/mL. Serial troponin levels showed elevated levels that were down-trending, from 42 ng/L on 23 October 2022 to 20 ng/L on 27 October 2022. An EKG on 25 October 2022 showed sinus tachycardia, with occasional premature ventricular complexes, with poor precordial lead R wave progression in V1–V3 (Figure 2). A cardiac catheterization was performed on 28 October 2022, which showed normal coronary arteries and a low–normal ejection fraction of 50–55% (Figure 3). A differential diagnosis of Takutsobo’s cardiomyopathy versus COVID-19 cardiomyopathy was made. Takutsobo’s cardiomyopathy was included in the differential diagnosis given days before, when the patient had been told that she may have metastatic breast cancer.

The patient was started on goal-directed medical therapy for heart failure while in the hospital, which included ramipril, spironolactone and bisoprolol. Ramipril was eventually discontinued during her hospitalization and, after an appropriate washout period, sacubitril/valsartan was started. Upon discharge, the patient was prescribed aspirin 81 mg, bisoprolol 20 mg, spironolactone 25 mg, atorvastatin 40 mg, sacubitril/valsartan 50 mg and dapagliflozin 5 mg.

The neurology team at UUH was consulted to evaluate for the possible cause of the patient’s syncope, intractable nausea and vomiting and fluctuating levels of consciousness. An MRI of the brain showed no acute lesions, and the patient remained afebrile with a normal white blood cell count. An EEG was conducted on 27 October 2022, which showed a slow posterior dominant rhythm, intermittent rhythmic slow and generalized EEG activity and a triphasic wave. These findings were supportive of a diagnosis of mild to moderate encephalopathy. Cerebrospinal fluid (CSF) analysis was initially deferred by the patient, and, due to the patient’s improving cognition, the acute onset of symptoms and the lack of differentiating characteristics between COVID-19-induced encephalitis and other viral encephalopathies, the neurology consult team then deemed it unnecessary to perform a CSF analysis. Throughout her hospitalization at UUH, the patient’s cognition improved; however, she continued to display intractable nausea and vomiting, with an inability to tolerate oral intake. Because of this, she was started on a percutaneous endoscopic gastrostomy tube, which was kept upon discharge. As a result of the patient’s improving mental status and lack of syncopal episodes over the week of admission, COVID-19 was believed to be the reason for her acute encephalopathy, rather than any autoimmune syndrome related to the patient’s metastatic breast cancer. High-dose corticosteroids were withheld due to the patient’s risk of acquiring a potentially lethal hospital-acquired infection. The patient was discharged to a rehabilitation facility. Sadly, in December 2022, the patient underwent an ultrasound-guided biopsy of the right breast and was diagnosed with invasive lobular carcinoma. 

## 3. Discussion

Based on phylogenetic analysis, SARS-CoV-2 falls under the Betacoronavirus genus of enveloped, positive-sense RNA viruses [21,22,23,24]. SARS-CoV-2 utilizes the angiotensin-converting enzyme 2 (ACE2) receptor to facilitate receptor-mediated endocytosis into mammalian host cells. Single-cell RNA sequencing has confirmed that type II alveolar cells, AT2 and myocardial cells express ACE2, making the lungs and heart primary targets for SARS-CoV-2 [25].

Patients at increased risk for COVID-19-associated myocarditis include the same demographic of patients at risk for severe forms of COVID-19 infection, including the elderly and patients with underlying conditions such as diabetes, cardiovascular disease and pulmonary diseases [26]. Based on pooled data, roughly 58% of patients diagnosed with COVID-19-induced myocarditis had one or more of the following conditions: hypertension, diabetes, obesity or asthma/chronic obstructive pulmonary disease [27]. In some severe cases of COVID-19, a shock state precedes the development of myocarditis, which could be an important differentiation from other forms of severe COVID-19, with studies showing 52% of patients with COVID-19 myocarditis who were in shock [28]. Diagnosis in nearly half of COVID-19 myocarditis cases was reached through the use of an echocardiogram. Findings included a reduced ejection fraction or dilated cardiomyopathy [29]. In some rare cases, physicians used cardiac catheterization techniques to rule out ischemic causes of presenting symptoms [30]. A separate etiology has oftentimes been linked to acute heart failure in COVID-19 patients, known as Takotsubo cardiomyopathy. Sympathetic overdrive related to emotional or physical stressors can lead to the transient apical ballooning of the left ventricle in patients [31]. Differentiation between Takotsubo cardiomyopathy and COVID-19 myocarditis can be made through MRI, which would show enhancement in the myocardium of patients experiencing COVID-19 myocarditis.

As the pandemic progressed, the incidence of local and systemic side effects from COVID-19 and its vaccination has increased. Cardiac conditions have been linked to both infection with SARS-CoV-2 and the vaccines produced to provide immunity against the virus [9,10,32]. We suspect that our patient developed viral myocarditis after an acute infection of SARS-CoV-2. However, an adverse reaction to the vaccine was possible because the patient completed the primary series for the mRNA vaccine within the past year. Oster et al. found that myocarditis secondary to vaccination occurred predominantly in male adolescents aged 12–24 and had an approximately 0.001% rate of incidence amongst the population [32]. Vaccine-induced myocarditis presented with similar characteristics of elevated troponin levels and abnormal EKG results [32]. We ruled out an adverse vaccine reaction because of the timing of onset of the cardiac condition and positive COVID-19 tests. COVID-19 vaccine myocarditis is believed to result from overly exuberant immune system activation and typically appears around 3 days after vaccination [33]. It can occur in persons with underlying cancer, but was not likely in this patient, whose vaccine was given 10 months prior to hospitalization [34].

Similar to COVID-19 myocarditis, COVID-19 encephalopathy is still a mostly unknown and burgeoning area of research regarding its exact causes, prevalence and long-term effects. The prevalence of COVID-19 encephalopathy is underestimated due to the lack of neurological screening in COVID-19-infected patients [35].

Post-concussion syndrome is a set of symptoms that can occur after a person has had a concussion. Symptoms can include headaches, dizziness, fatigue, irritability, anxiety and difficulty with memory and concentration. In some cases, these symptoms can persist for weeks or even months after the initial injury [36]. The patient presented with post-concussion syndrome months prior to her COVID-19 diagnosis. In the case of post-concussion syndrome, the symptoms of encephalopathy could have been caused by the brain injury that occurred during the initial concussion or more acutely by COVID-19. Prior brain injury from concussions could have led to the compounded display of symptoms in this patient [36].

Paraneoplastic conditions associated with the patient’s finding of invasive lobular carcinoma were explored as a possible reason for her development of encephalitis. CSF findings favoring a diagnosis of paraneoplastic syndrome include oligoclonal IgG, lymphocytic pleocytosis and elevated protein [37]. A rare case of paraneoplastic encephalitis has been described by Agrawal, in which the patient presented with a 3-month history of consistent neurologic dysfunction, described as a sensation of pins and needles in her legs, balance problems and cognitive decline, and was later revealed to have invasive ductal carcinoma in the right breast [38]. CSF from the Agrawal case showed elevated oligoclonal bands and lymphocytic pleocytosis, but with normal protein. The Agrawal patient thus exhibited two of the three findings often associated with paraneoplastic syndrome. In our case, the presentation of neurologic symptoms coincided with COVID-19 infection and was even more acute in comparison to the case by Agrawal. Lumbar puncture was deferred by our patient upon arrival at UUH. However, CSF data could have helped to better differentiate COVID-19 encephalitis from paraneoplastic encephalitis. Clinical judgement and patient presentation pointed towards a diagnosis of viral encephalitis over paraneoplastic encephalitis.

The patient’s immunocompromised state due to cancer and/or cancer treatment put her at greater risk than the general population for severe COVID-19 infection and complications stemming from infection [39]. The reporting of cardiovascular complications stemming from COVID-19 in cancer patients has been limited. Many cancer patients are treated with immune checkpoint inhibitors (ICI), which have been known to be associated with myocarditis; however, this patient’s chemotherapy regimen did not include any ICIs [40]. This patient’s cancer treatment was held until she made a full recovery from COVID-19 and associated complications. However, the subsequently confirmed metastatic breast cancer may have made her more vulnerable to developing this uniquely complicated case of COVID-19. It may also have made her response to the vaccine less robust [40]. It is also possible that the COVID-19 infection was simply coincident with the encephalitis and myocarditis. However, myocarditis is not generally associated with breast cancer or paraneoplastic syndromes, although it may be associated with chemotherapeutic agents [41]. This was a complicated case for which we do not have definitive answers as to the specific cause of each of the manifestations. Further research is needed into the cardiovascular and CNS risks posed to immunocompromised patients with severe COVID-19.

## Figures and Tables

**Figure 1 vaccines-11-00541-f001:**
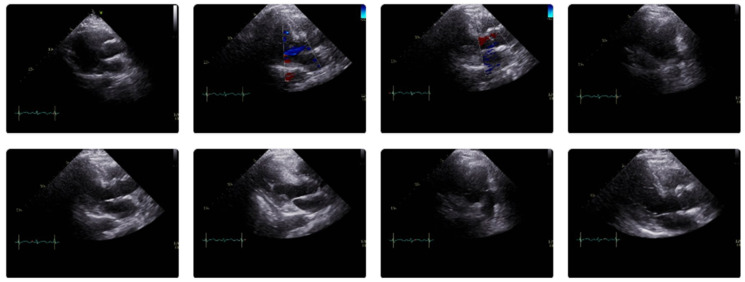
Echocardiogram from 26 October 2022, which showed an ejection fraction of 31% with wall motion abnormalities.

**Figure 2 vaccines-11-00541-f002:**
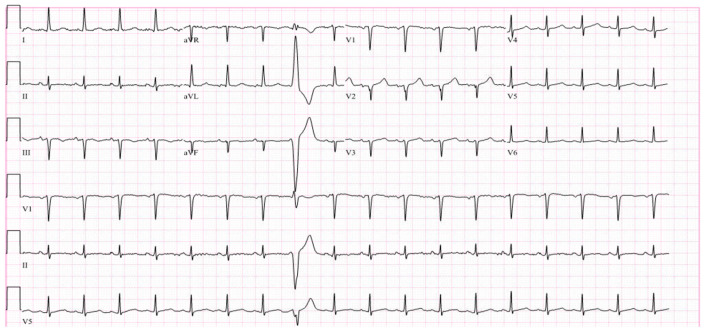
EKG showing premature ventricular complexes and poor precordial lead R wave progression.

**Figure 3 vaccines-11-00541-f003:**
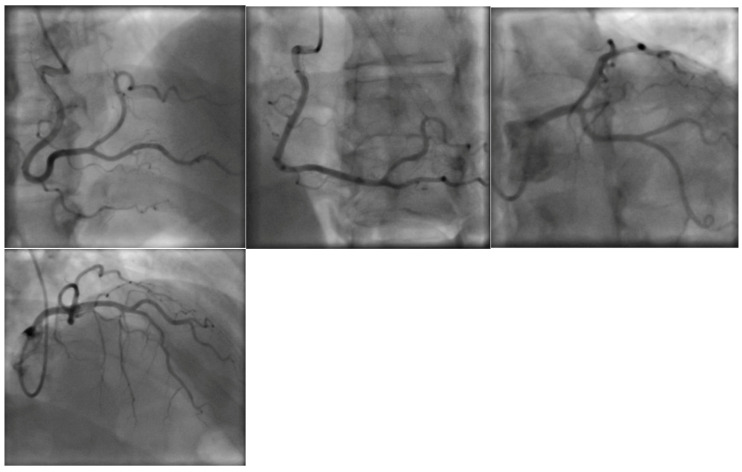
Cardiac catheterization performed on 28 October 2022, which showed normal coronary arteries and a low–normal ejection fraction of 50–55%.

**Table 1 vaccines-11-00541-t001:** Patient lab results and reference ranges.

Lab	Date	Value	Normal Value
Troponin T	23 October 2022	42 ng/L	<14 ng/L
	27 October 2022	27 ng/L	
CA 15-3	24 October 2022	16.8 U/mL	<30 U/mL
CA 19-9	24 October 2022	7 U/mL	<35 U/mL
CA 27.29	24 October 2022	21.9 U/mL	<38.6 U/mL
CEA	24 October 2022	1.5 ng/mL	<3.4 ng/mL
proBNP	27 October 2022	7616 pg/mL	<125 pg/mL

**Table 2 vaccines-11-00541-t002:** Results of echocardiogram from 26 October 2022.

Anatomy	Label	Value	Normal Value
Interventricular Septum	IVSd	0.8 cm	(0.6–0.9 cm)
Left Atrium	LADs	2.3 cm	(2.7–3.9 cm)
Left Ventricle	LVOTd	1.7 cm	(1.8–2 cm)
Left Ventricle	LVDd	3.3 cm	(3.8–5.2 cm)
Left Ventricle	LVEF, MOD4	37%	(54–74%)
Left Ventricle	LVEF, MOD2	25%	(54–74%)
Left Ventricle	LV Mass Index	41.5 g/m^2^	(43–95 g/m^2^)

## Data Availability

All relevant data are provided in the paper.

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
