# Peer review of "A Rare Single Case of COVID-19-Induced Acute Myocarditis and Encephalopathy Presenting Simultaneously"

_vaccines, 2023, doi:10.3390/vaccines11030541_

Round 1
Reviewer 1 Report
The article is very intersting however there are some comments to be addressed by the authors
1. The title may be altered to reflect the content and indicate that it is just one case report . COVID-19 induced acute myocarditis and encephalopathy may give an impression of the generalized effect of COVID-19
2. Authors should elborate in the discussion the acute myocarditis and encephalopathy relattion to both infection and COVID-19 vaccine administeration.
Author Response
We thank this reviewer for thoroughly scrutinizing our manuscript. As requested, we have revised the manuscript and addressed the specific comments of this reviewer. The revised sections are delineated in red in a marked copy of the manuscript text.
Comments of Reviewer 1 with responses:
Comment 1. The title may be altered to reflect the content and indicate that it is just one case report . COVID-19 induced acute myocarditis and encephalopathy may give an impression of the generalized effect of COVID-19
Response: We have changed the title to: “A rare single case of Covid-19 induced acute myocarditis and encephalopathy presenting simultaneously”
Comment 2. Authors should elaborate in the discussion the acute myocarditis and encephalopathy relation to both infection and COVID-19 vaccine administration.
Response: We appreciate the feedback about elaborating more on the relation between infection and vaccine administration on acute myocarditis and encephalopathy. Unfortunately, in this case, we could not distinguish vaccine from infection other than to note in lines 101-106 that the patient had waning immunity despite prior vaccination and infection. We have inserted more discussion (lines 197-200 and 214-229).
Reviewer 2 Report
The authors find myocarditis and encephalopathy in a 54 years old female who had three COVID-19 vaccinations and was tested positive for COVID-19 within one year of the last vaccination. Importantly, the woman also had newly diagnosed breast-cancer with widespread filiarization. The authors believe that this is the first case of concomitant mycarditis and encephalopathy related to COVID-19 acute infection or possibly vaccination.
Major concerns:
1. Was there a CSF examination for infectious or immunological causes of encephalopathy?
2. Did the authors look for autoimmune encephalopathy related to breast cancer?
3. Did the authors look for COVID-19 antibodies?
4. Were plasmapheresis and high dose corticosteriods considered for potential immunological causes?
5. Conclusions must be altered if immunological causes not related to COVID-19 are possible.
Author Response
We thank this reviewer for thoroughly scrutinizing our manuscript. As requested, we have revised the manuscript and addressed the specific comments of this reviewer. The revised sections are delineated in red in a marked copy of the manuscript text.
Comments of Reviewer 2 with responses:
Comment 1. Was there a CSF examination for infectious or immunological causes of encephalopathy?
Response: We have now noted in the paper that the patient made the choice to defer CSF analysis and the neurology consult team then deemed it unnecessary to do a CSF analysis based on the patient’s acute presentation and quick recovery time from the encephalopathic condition. Lines 145-149.
Comment 2. Did the authors look for autoimmune encephalopathy related to breast cancer?
Response: We have now elaborated further on this within both the case report section (lines 152-156) as well as the discussion section (lines 214-229). The rarity of paraneoplastic symptoms related to invasive lobular carcinoma of the breast was discussed, with a corresponding case report delineating the differences between viral encephalitis and autoimmune encephalitis related to breast cancer
Comment 3. Did the authors look for COVID-19 antibodies?
Response: The patients antibody titer reading is now included. It shows a low yield of Covid-19 antibodies in this doubled vaccinated and boosted patient. Lines 101-106.
Comment 4. Were plasmapheresis and high dose corticosteroids considered for potential immunological causes?
Response: Based on medical advice and low reason to suspect autoimmune encephalitis, high dose corticosteroids and plasmapheresis were withheld from this patient due to increased risk of developing a hospital acquired infection (lines 155-156).
Comment 5. Conclusions must be altered if immunological causes not related to COVID-19 are possible.
Response: Conclusion has been altered to more thoroughly discuss a rare case of paraneoplastic encephalitis associated with breast cancer patients and how our patient’s presenting symptoms differentiated from that of the patient discussed in the other case report. We make clear in the conclusion that the case is complicated and that we cannot trace each manifestation to a specific cause (lines 214-229 and 239-244).
Round 2
Reviewer 2 Report
none